# Tumor Tropism of DNA Viruses for Oncolytic Virotherapy

**DOI:** 10.3390/v15112262

**Published:** 2023-11-16

**Authors:** Junior A. Enow, Hummad I. Sheikh, Masmudur M. Rahman

**Affiliations:** 1Biodesign Center for Personalized Diagnostics, Biodesign Institute, Arizona State University, Tempe, AZ 85287, USA; 2School of Life Sciences, Arizona State University, Tempe, AZ 85287, USA

**Keywords:** oncolytic virus, oncolytic virotherapy, DNA virus, tumor tropism, poxvirus, herpesvirus, adenovirus

## Abstract

Oncolytic viruses (OVs) have emerged as one of the most promising cancer immunotherapy agents that selectively target and kill cancer cells while sparing normal cells. OVs are from diverse families of viruses and can possess either a DNA or an RNA genome. These viruses also have either a natural or engineered tropism for cancer cells. Oncolytic DNA viruses have the additional advantage of a stable genome and multiple-transgene insertion capability without compromising infection or replication. Herpes simplex virus 1 (HSV-1), a member of the oncolytic DNA viruses, has been approved for the treatment of cancers. This success with HSV-1 was achievable by introducing multiple genetic modifications within the virus to enhance cancer selectivity and reduce the toxicity to healthy cells. Here, we review the natural characteristics of and genetically engineered changes in selected DNA viruses that enhance the tumor tropism of these oncolytic viruses.

## 1. Introduction

Owing to their multifaceted antitumor effects, oncolytic viruses (OVs) have recently gained prominence as innovative agents for cancer immunotherapy. These viruses belong to various families, selectively targeting and destroying cancer cells while leaving normal cells unharmed. Oncolytic viruses either naturally target cancer cells or have been engineered to do so. However, all OVs are expected to induce tumor cell lysis by replication or other mechanisms, spread to the neighboring cells, and influence local and systemic antitumor immunity for tumor clearance.

OVs are recognized as one of the emerging cancer treatment modalities that can complement most existing cancer therapies, including radiation, chemotherapy, surgery, immunotherapy, and cell-based treatments. An attribute of OVs that makes them suitable immunotherapy options is the ability to engineer OVs with transgenes that can alter the tumor microenvironment by expressing cytokines, chemokines, immune checkpoint inhibitors (ICIs), or molecules that can directly lyse the cancer cells.

Foremost among all characteristics, the pivotal feature of every OV is its ability to selectively infect and replicate within tumors. This cancer tropism, often referred to as oncotropism, in OVs has been accomplished through various approaches: (i) engineering OVs to specifically target cancer cells, either in the entry stage or during post-entry replication steps; (ii) utilizing IFN-sensitive viruses that cannot replicate in normal cells but thrive in cancer cells with defective IFN pathways; (iii) crafting OVs devoid of one or more immune-evasion genes; and (iv) harnessing the unique metabolic pathways of cancer cells. All these approaches provide safety features in OVs for clinical application. DNA viruses that have demonstrated promise in cancer therapy include adenoviruses, parvoviruses, and various poxviruses like vaccinia virus (VACV) and myxoma virus (MYXV), as well as herpesviruses. The large DNA genomes of poxviruses and herpesviruses offer the potential for sizable foreign genetic inserts. The ability to genetically modify these viruses not only enables enhanced tropism for cancer cells but also significantly reduces their ability to infect normal cells, thereby bolstering the safety of these oncolytic viruses (Figure 1). This review focuses on the cellular and tumor tropism of selected oncolytic DNA viruses that have been well studied and engineered for tumor-selective replication and enhanced oncolytic activity (Table 1).

## 2. Oncolytic Poxviruses

### 2.1. Natural Tropism of Poxviruses for Cancer Cells

The Poxviridae family of viruses comprises large (~400 nm) enveloped viruses with a double-stranded DNA genome of 130 to 375 kbp. Poxviruses are divided into two subfamilies, namely, Chordopoxvirinae, which infect vertebrates, and Entomopoxvirinae, which infect invertebrates. The Chordopoxvirinae subfamily is divided into 18 genera based on their unique serological reactions. Because of their selected host tropism, poxviruses from different genera have been tested for oncolytic activity. Representative poxviruses from other genera that have been tested for oncolytic activity are from *Orthopoxvirus,* including vaccinia virus (VACV), raccoon poxvirus (RCNV), and cowpox virus (CPXV); *Leporipoxviruses*, including myxoma virus (MYXV); *Yatapoxvirus*, including Tanapoxvirus (TPV) and Yaba-like disease viruses (YLDVs); *Parapoxvirus*, including Orf virus; and *Avipoxvirus*, including fowlpox virus and Canarypox. These representative poxviruses from different genera have been genetically engineered for tumor-specific replication and activation of antitumor immune responses. Depending on their ability to make progeny virions after completing the replication cycle, oncolytic poxviruses can be broadly categorized into replicating poxviruses, non-replicating vaccine vectors, and chimeric poxviruses. Replicating poxviruses can be further subdivided into attenuated (reduced pathogenicity with specific gene deletions) and non-attenuated (unmodified) viruses. Non-replicating poxviruses, for example, modified vaccinia virus Ankara (MVA), are mainly developed as vaccine platforms. However, in oncolytic virotherapy, they have been exploited as vectors for the expression of therapeutic transgenes. Chimeric poxviruses have recently emerged as novel oncolytic viruses which can be generated by co-infecting more than two poxviruses from the same or different genera.

Poxviruses are unique among DNA viruses because they replicate entirely in the cytoplasm of infected cells using their own viral replication machinery. Although poxviruses are host-specific and cause disease in only selected host species, most can bind and enter diverse cell types, including normal and cancerous cells. However, normal cells can resist poxvirus replication and generation of progeny virions because of functional innate antiviral responses. On the other hand, almost all cancer cell types are defective in some part of the antiviral signaling pathways, allowing oncolytic poxviruses from different genera to infect and replicate in cancer cells originating from species outside their pathogenic hosts [1]. For example, MYXV, a rabbit-specific poxvirus that causes a lethal disease called myxomatosis in European rabbits, can infect and replicate in cancer cells originating from humans, mice, canines, and other species [2,3]. Apart from innate immune antiviral pathways, the cell surface proteins and differentiation or activation stage of primary cells can also prevent poxvirus binding and entry into the primary cells. For example, studies with VACV demonstrated that monocytes show maximum infection among different human primary hematolymphoid populations, followed by B lymphocytes, DCs, and NK cells, and T cells are the least infected [4,5]. However, activated B and T cells are more susceptible to VACV binding and infection [4,6]. Activated B and T cells are also more vulnerable to MYXV, which can only bind to nonactivated primary human T cells. Still, upon activation with anti-CD3/CD28 or non-specific T-cell activators, the bound MYXV can infect those cells and complete its replication cycle [7]. Notably, neither MYXV nor VACV can bind to primary human CD34+ stem cells [8,9]. These results suggest that for poxviruses, infection of normal primary cells is determined by cell type and “state”. Also, normal cell susceptibility to poxviruses is determined at multiple levels of virus binding, entry, replication, and innate immune signaling pathways, many of which cancer cells are missing.

The process of poxvirus binding, entry, infection, and replication is a multistep sequence involving the attachment, internalization, and release of virion cores into the cytoplasm. This process relies on at least four virus-encoded proteins—D8, A27, H3, and A26—which play pivotal roles in poxvirus cell attachment. Twelve proteins are involved in forming the entry–fusion complex (EFC), which mediates the internalization of virions into the cytoplasm of cells [10,11]. Although these proteins are mostly conserved among different genera of poxviruses, their ability to replicate successfully in a given cell type largely depends on the modulation of diverse intracellular signaling pathways.

### 2.2. Vaccinia Virus and Other Orthopoxviruses

VACV is the prototype member of the genus orthopoxviruses. Vaccinia has been the primary focus of most studies on poxvirus biology and pathogenesis, largely because of its use as a smallpox vaccine. Also, VACV has been developed as a vaccine vector against many infectious diseases and as an oncolytic virus [12]. Several strains of VACV, for example, Lister, Wyeth, and Western Reserve (WR), have been explored as oncolytic viruses [13]. In infected cells, VACV produces multiple forms of infectious particles, namely, intracellular mature virion (IMV), intracellular enveloped virion (IEV), cell-associated enveloped virion (CEV), and extracellular enveloped virion (EEV) [14]. Of all these various forms, single-layered intracellular IMVs are the most abundant and are released following cell lysis. These released IMVs can efficiently infect neighboring cells through the process of endocytosis [10]. A minority of IMVs is enveloped by the double-layer Golgi membrane, forming IEVs, which are then transported to the cell periphery to become CEVs. A smaller fraction of IMVs directly moves to the cell periphery, fuses with the cell membrane, acquires an outer envelope, and is released as EEVs. VACV’s EEV has evolved to facilitate rapid systemic spread within the host and evade immune-mediated clearance. EEV-enhanced VACV strains displayed improved tumor spread after systemic delivery, resulting in improved antitumor effects [15]. The VACV EEV strain is resistant to complement activation owing to the incorporation of host regulators of complement activation (RCA) proteins, including CD46, CD55, CD59, CD71, CD81, and MHC-I, into its envelope [16].

#### 2.2.1. Enhancing Tumor-Selective Replication of VACV

To further improve the oncolytic properties of VACV EEV, several approaches have been used to enhance virus spread in the tumor bed. Mutation of lysine 151 to glutamic acid (A34 K151E substitution) of viral A34 protein, which is involved in the induction of actin tails and the release of EEV from the surface of infected cells, led to an increase in virus spread and the formation of EEV [15]. However, further deletion of B18R, which encodes soluble interferon receptor (IFNR) and suppresses host IFN, increased IFN’s antitumor activity.

Previous studies demonstrated that deleting B18R and expressing IFN-β resulted in a virus that cannot infect normal cells but can replicate within cancer cells, enhancing tumor killing [17]. Another EEV-associated protein, B5R, responsible for viral morphogenesis, trafficking, and dissemination and involved in the non-specific targeting of healthy and tumor cells, was engineered to enable tumor-specific replication, evading virus neutralization while retaining oncolytic activity [17,18].

As mentioned previously, poxviruses, particularly various strains of VACVs, naturally possess the ability to infect tumor cells. Genetic modifications are introduced to improve VACV’s selective replication in tumor cells and protect normal cells. An attenuated VACV WR strain with two deletions (vvDD) was developed to selectively target and eliminate cancer cells [19]. Vaccinia growth factor (VGF) and thymidine kinase (TK) were the deleted genes. Deleting TK resulted in significant attenuation in normal, non-dividing cells while retaining replication ability in tumor cells, which were actively dividing [20]. VGF, an extracellular viral growth factor, is required for virus replication in neighboring healthy cells [21]. Thus, the deletion of VGF stopped virus infection of healthy cells outside the tumor bed. In summary, the deletion of both TK and VGF produced an oncolytic VACV with a high degree of tumor selectivity. Subsequently, TK and VGF deletions have been introduced into other oncolytic VACV strains and orthopoxviruses [21,22]. Apart from VGF and TK genes, additional genes in VACV were deleted to create more attenuated and tumor-targeted OVs. For example, the VACV Lister strain was used to create GLV-1H68, which has the deletion of three genes [23]. In addition to TK (J2R) gene deletion, F14.5L, which encodes a secretory signal peptide, and A56R (encoding hemagglutinin) were deleted in GLV-1H68. The deletion of F14.5L and A56R further attenuated the virus but retained tumor-specific replication. Subsequently, these genes were deleted in other VACV strains [19,20]. Besides VACV, TK was also removed from other orthopoxviruses like CPXV and raccoon poxvirus to achieve tumor-selective replication [22,24].

To further weaken VACV’s activity in normal cells while maintaining replication in tumor cells, both VGF and O1 genes were removed [25]. VGF acts in a paracrine and autocrine manner to enhance virus replication by activating the EGFR-dependent mitogen-activated protein kinase (MAPK)–ERK pathway [26]. The O1 protein is essential for sustaining the activation of extracellular signal-regulated kinase 1/2 (ERK1/2) signaling initiated by VGF [27]. In tumor cells, oncogenic mutations in the EGFR, RAS, and Raf genes cause constitutive ERK1/2 activation in the MAPK pathway. Hence, the deletion of both VGF and O1 enabled MAPK-dependent tumor-specific replication of VACV while sparing normal cells [25]. The effectiveness of this virus was assessed in a xenograft mouse model of pancreatic ductal adenocarcinoma. In addition to eradicating cancer cells, this oncolytic VACV can infect and eliminate tumor vascular endothelial cells [28].

The attenuation of virus replication in normal cells is critical for generating safer OVs. In addition to TK (encoded by J2R), another viral protein encoded by the F4L gene, a ribonucleotide reductase, is also involved in nucleotide synthesis. The F4L-deleted virus showed antitumor immunity and a higher safety profile in cancer models. Moreover, another modified VACV with the deletion of four viral genes (A48R, B18R, C11R, and J2R) involved in various pathways like metabolism, proliferation, and signaling demonstrated antitumor activity while maintaining tumor selectivity in vivo [29].

#### 2.2.2. Deletion of Immune-Evasion Genes in VACV to Enhance Tumor Selectivity and Activation of Antitumor Immune Response

A recent study demonstrated that deleting three key immune-evasion gene products (C10L, N2L, and C6L) from VACV preserved its replication ability in cancer cells [30]. Proteins encoded by VACV, including C10, A46, N2L, and C6, acted as antagonists in the TLR3-IRF3 signaling pathway in various stages [31]. The deletion of these genes facilitated the phosphorylation of IRF3, a pivotal protein in the activation of TLR3, leading to heightened cytotoxic T lymphocyte (CTL) responses in a syngeneic mouse model. Another study adopted a similar approach by deleting multiple immunomodulatory genes (N1L, K1L, K3L, A46R, and A52R). Again, the goal was to enhance tumor-selective replication and attenuation in non-tumor cells [32]. In vitro, the deletion of independent genes resulted in similar or improved in vitro replication. However, in vivo studies demonstrated enhanced oncolytic activity, particularly for viruses lacking the K1L, A46R, and A52R genes. These deletions facilitated potent antitumor immunity through modulation of the tumor microenvironment (TME).

The deletion of immune-evasion genes can activate antitumor immune responses through virus-induced cytokine production. As an example, the oncolytic VACV construct with deleted N1L protein expression (VVΔTKΔN1L) exhibited enhanced virus-induced cytokine production, resulting in an increase in the number of circulating NK cells in surgery-induced metastatic models of cancer [33]. Moreover, the expression of the cytokine IL-12 within this N1LKO background (VVΔTKΔN1L-IL12) significantly improved efficacy in the head and neck cancer model [33]. VACV N1L plays a crucial role in immune evasion and is indispensable for poxvirus virulence [34,35]. N1L inhibited NF-kB signaling, and the deletion of N1L enhanced NK cell responses to viral infection and improved the generation of immediate and long-term memory CD8+ T-cell responses [36,37]. Thus, the deletion of immunomodulatory genes can enhance the tumor-specific tropism and oncolytic activity of poxviruses.

### 2.3. Other Poxviruses with Natural Tropism for Cancer Cells

Yatapoxviruses, including Tanapoxvirus (TPV) and Yaba-like disease virus (YLDV), have been harnessed for oncolytic virotherapy. TPV and YLDV are two monkey viruses that cause human infection [38]. While these viruses typically result in a relatively benign human infection, they can infect and eliminate cancer cells. The cellular tropism of TPV was examined using primary human dermal fibroblasts (pHDFs) and peripheral blood mononuclear cells (PBMCs) [39]. pHDFs supported TPV infection and replication. Upon TPV infection of PBMCs, monocytes (CD14+) constituted the predominant infected population. Nevertheless, the virus did not replicate when monocytes were differentiated into macrophages [39]. In vitro, TPV can replicate within various human cancer cell lines [40]. TPV can replicate in human xenograft tumors in nude mice. Like other poxviruses, TK (66R) was deleted from TPV to replicate in tumors selectively. The deletion of TPV-encoded neuregulin (NRG), an EGF-like growth factor (encoded by 15L) that enhances various types of cell proliferation, as seen in TPVΔ15L, resulted in enhanced regression of melanoma tumors compared to wild-type TPV [41]. Additionally, the deletion of the viral type I IFN binding receptor (136R) enables selective replication in tumors and abortive replication in primary cells responsive to IFN. In contrast to TPV, the oncolytic potential of YLDV has been evaluated in a limited number of studies [42].

Orf virus (ORFV) is a prototype member of the genus Parapoxvirus that causes benign skin disease in its natural ungulate host. Like many other host-restricted poxviruses, ORFV also showed oncolytic activity in vitro and in vivo [43,44]. Screening using Parapoxvirus ovis (strain D1701) revealed antitumor activity in human xenograft models and syngeneic B16-F10 melanoma models [45].The oncolytic activity of Parapoxvirus is mainly mediated by NK cells and the induction of the Th1 immune response. Oncolytic MYXV, a member of Leporipoxvirus, exhibits a natural affinity for cancer cells and has undergone testing for oncolytic activity in various preclinical cancer models. A recent MYXV review highlighted different oncolytic MYXV aspects not described in this review [3].

### 2.4. Novel Chimeric Poxviruses with Enhanced Tropism for Cancer Cells

In recent years, chimeric poxviruses have been generated by co-infecting more than two viruses from the same or different genera. The reason is to enhance the tumor-selective replication and the killing ability of poxviruses. A novel chimeric Parapoxvirus CF189 was isolated by co-infection of MDBK cells with Orf virus strain NZ2 and pseudocowpox virus strain TJS [46]. From the more than 100 isolates generated from this co-infection, isolate CF189 was selected because of the demonstration of its potent cell-killing ability in human triple-negative breast cancer cell lines. However, further studies are needed to understand the mechanisms that facilitated enhanced cancer-cell killing.

A chimeric orthopoxvirus was created by co-infecting CV-1 cells with multiple (nine) orthopoxvirus strains, including cowpox virus strain Brighton, raccoon poxvirus strain Herman, rabbitpox virus strain Utrecht, and various VACV strains: Western Reserve (WR), International Health Department (IHD), Elstree, CL, Lederle-Chorioallantoic (LC), and AS [47]. Out of over 100 isolates, CF33 and CF17 were chosen because of their exceptional ability to kill cancer cell lines from the NCI-60 panel. These isolates were also assessed against human pancreatic cancer and TNBC cell lines. Later, it was shown that endogenous Akt, also known as protein kinase B (PKB), activity in TNBC cells regulates the replication of these isolates, as shown before with VACV and MYXV [48]. Likewise, CF17 exhibited improved cell-killing abilities and effectiveness in the intraperitoneal (IP) ovarian cancer model [49] Genome sequencing of CF33 indicated that the virus genome is primarily derived from three VACV strains—IHD, WR, and Lister—with no sequences detected from the raccoon poxvirus genome [50]. A similar method was employed to create the chimeric poxvirus deVV5, achieved through the recombination of four VACV strains: WR, Wyeth, MVA, and Copenhagen [51]. deVV5 showed enhanced cancer cell killing and tumor selectivity.

Moreover, the deletion of TK from deVV5 and the incorporation of a suicide gene FCU1 (resulting in VV5-fcu1) led to attenuation in normal cells. Nevertheless, despite the creation of these chimeric viruses and the selection of recombinant virus isolates with the highest replication capability and cytotoxicity against cancer cells, these viruses effectively replicate and eliminate only a limited subset of available cancer-cell lines specific to a particular cancer type [50]. These findings indicate that, instead of implementing extensive alterations in the viral genome, the knockout or insertion of specific viral genes to enhance cancer cell tropism and cytotoxicity remains a valuable strategy.
viruses-15-02262-t001_Table 1Table 1List of selected modified oncolytic DNA viruses and their tumor tropism.FamilyVirusNatural HostMutant NamesTargeting PropertiesEngineered TropismIntroduced TransgeneTarget Cancer TypeRef.Poxviridae Vaccinia virus UnknownVACV-EEV Enhanced virus spread in tumor beds. Viral inhibition of complement. Lack of IFN inhibition. K151E mutation of viral protein A34 (A34 K151E) and B18R deletion.N/APC3 human prostate cancer [15]
Vaccinia virus WR strain UnknownvvDDAttenuate infection of healthy cells but retain replicative ability in dividing cancer cells. Viral thymidine kinase and vaccinia growth factor (VGF) gene deletion. N/AHuman and murine adenocarcinoma cell lines[20,21]
Vaccinia virus Lister strainUnknown GLV-1H68Attenuate infection of healthy cells but retain tumor-selective replication. Deletions of viral thymidine kinase (J2R), secretory signal peptide (F14.5L), and hemagglutinin (A56R) genes. N/A GI-101A human breast tumors xenograft mouse model[23]
Vaccinia virus LC16mO strainUnknownMDRVVTumor-selective virus for MAPK–ERK pathway. Deletions of VGF and O1 genes. N/AHuman pancreatic ductal adenocarcinoma xenograft mouse model[25,26,27]
Vaccinia virus WR strainUnknown∆*F4L*∆*J2R*
VACVSelective targeting of cancer cells because of the high level of ribonucleotide reductase enzyme expression that the virus lacks. Deletions of viral thymidine kinase (J2R) and F4L genes.N/A Xenograft human RT112-luc orthotopic bladder cancer model[52]
Vaccinia virus WR strainUnknownWR-Δ4 Targeting of viral genes that act on metabolic, proliferation, and signaling pathways to confine viral tropism to cancer cells. Deletions of viral thymidine kinase (J2R), soluble interferon receptor (B18R), vaccinia growth factor (C11R), and thymidine kinase (A48R) genes. N/A Mouse B16F10 syngeneic melanoma model[29]
Vaccinia virus WR/TK strainUnknown WR/TK−/3ΔEnhanced tumor selectivity both in vitro and in vivo and enhanced IRF3 phosphorylation within the cancer cells. Deletion of immune-evasion genes (C10L, N2L, C6L) that antagonize the TLR3-IRF3 pathway at different levels. N/A Mouse syngeneic Renca (kidney derived) tumor model[30,31]
Vaccinia virus WR strain F13L+Unknown ΔN1L VV, ΔK1L VV, ΔK3L VV, ΔA46R VV, and ΔA52R VVEffectively target and kill tumor cells by diminishing the viral antiviral properties. Deleting the N1L, K1L, K3L, A46R, and A52R genes increased antitumor immune response and heightened production of virus-induced cytokines.N/AHuman colorectal adenocarcinoma cell line DLD-1, human ovarian cancer cell line A2780[32]
Vaccinia virus WR strainUnknownVVΔTKΔN1L-IL12Tumor-specific targeting by the virus, more outstanding viral-induced cytokine production, and elevated circulating NK-cell levels. N1L deletion correlated with greater levels of virus-induced cytokines and elevated NK-cell levels. IL12CT26 mouse metastatic colon adenocarcinoma and metastatic lung squamous cell carcinoma LLC. Surgical models of head and neck cancer in Syrian hamsters [33]
Tanapoxvirus Human TPVΔ15LCancer-specific targeting. TK (66L), neuregulin (NRG), and EGF-like growth factor (15L) have been deleted. N/AHuman melanoma xenograft model in nude mice [41]
Chimeric Parapoxvirus N/ACF189Robust tumor cell targeting by virus. Chimeric Parapoxviruses were constructed by co-infecting MDBK cells with Orf-virus strain NZ2 and pseudocowpox virus strain TJS. N/A Mouse models of human triple-negative breast cancer cell lines MDA-MB-468 [46]
Chimeric orthopoxvirus N/A CF33Superior ability to kill human pancreatic cancer cells. The chimeric orthopox virus was created by co-infecting nine strains of the orthopox virus. (see section on novel chimeric poxviruses for the different viruses used.) N/A Mouse xenograft models of human pancreatic ductal adenocarcinoma [47]
Chimeric vaccinia virus N/A deVV5Improved cancer-killing capacity and tumor selectivity in vitro. Chimera was created by co-infecting VACV strains Copenhagen, Western Reserve, Wyeth, and the attenuated VACV Ankara. The TK gene was deleted to attenuate virus replication in normal primary cells. N/AIn vitro screen of human cancer cell lines[51]HerpesviridaeHerpes simplex virus 1 Humans N/A Reduced tropism for neuronal cells and selective targeting of cancer cells. Deletion in viral protein ICP34.5 that is involved in inhibiting PKR. N/A The modified oHSV-1 is safe in Phase-I clinical trials in glioma and melanoma patients [53]
Herpes simplex virus 1HumansN/ATumor-selective replication due to high levels of ribonuclease reductase expression. The viral gene (UL39) for the ICP6 protein is deleted. N/A Deleted ICP6 oHSV is currently in Phase I clinical trial liver metastasis and primary liver cancer [54]
Herpes simplex virus 1HumansrRp450Enhanced tumor-selective targeting and minimal effect on normal cells. Deletion of *ICP6* gene.Introduction of rat cytochrome P450 2B1 that serves as a prodrug enzyme for cyclophosphamide. rRp450 showed therapeutic benefit in mouse models of diffuse colon cancer liver metastasis [55]
Herpes simplex virus 1HumansG47∆Enhanced MHC-I antigen presentation, enhanced cytopathic effect in-vitro and tumor-selective targeting. Deletion of *ICP6* gene, *ICP47* gene, and both copies of the *ICP34.5* gene.N/A G47∆ has shown therapeutic efficacy against mouse models of Neuro2a neuroblastoma tumors and human U87MG glioma tumors[56]
Herpes simplex virus 1HumansR-LM13 Engineered to target HER-2 expressing tumor cells selectively. Insertion of scFV targeting HER-2 into glycoprotein gD. N/A R-LM13 has demonstrated therapeutic efficacy against mice models of human ovarian cancer (SK-OV-3) and breast cancer (MDA-MB-453 and BT-474) [57,58,59]
Herpes simplex virus 1HumansrQNestin34.5The virus is engineered to replicate selectively under the Nestin promoter’s influence. Insertion of a Nestin-specific promoter within the *ICP34.5* gene.N/A rQNestin34.5 treatment increased the survival rate of nude mice by >90% of 77.8% mice bearing intracerebral human U87EGFR glioma[60]AdenoviridaeAdenovirus serotype 5 Human Ad5-3Δ-A20TAd5-3Δ-A20T is engineered to target αvβ6 integrin-expressing cells selectively. The Ad5 mutant expresses the αvβ6-binding peptide that can infect and kill αvβ6-expressing cells. αvβ6-binding peptideAd5-3Δ-A20T efficiently inhibited the growth of human pancreatic cancer xenograft murine models [61]
Adenovirus serotype 5HumanAdV5 hTERT AdV5 hTERT can selectively replicate in cells expressing high levels of telomerase activity (cancer cells). AdV5 expressing the E1A and E1B genes under the influence of an hTERT promoter. Human telomerase reverse transcriptase promoter. AdV5 hTERT intratumoral injection resulted in the growth inhibition of human lung tumor models[62]
Adenovirus serotype 5HumanONYX-15ONYX-15 can selectively replicate in tumors lacking a functional p53.This virus lacks the 55kDa E1B gene region.N/AONYX-15 resulted in antitumor effects in mouse–human tumor xenografts of cervical (C33A) and laryngeal (HLaC) carcinoma lacking functional p53. ONYX-15 is being used in clinical trials for head and neck cancer[63]
Adenovirus serotype 5Human Gendicine^®^Gendicine can selectively kill tumors by forcing them to undergo apoptosis. Gendicine lacks the E1 gene region. Introduction of the p53 protein driven by a Rous sarcoma virus promoterGendicine^®^ is approved to treat head and neck squamous cell carcinoma in China. [64,65]


## 3. Oncolytic Herpes Simplex Virus (oHSV)

Mammalian herpesviruses belong to the Herpesviridae family of viruses and are categorized into three subfamilies: alphaherpesviruses, betaherpesviruses, and gammaherpesviruses. Human alphaherpesviruses comprise HSV-1, HSV-2, and varicella-zoster virus (VZV). HSV-1 has gained recognition as an oncolytic virus (OV) following successful clinical trials against various cancer types. Two oncolytic HSV-1-derived viruses approved for cancer treatment are T-VEC (Talimogene Laherparepvec) and G47Δ [66].

Herpesviruses are enveloped virions with a substantial dsDNA genome of approximately 152 kb. In contrast to poxviruses, herpesviruses replicate within the nucleus of infected cells [67]. Every member of the Herpesviridae family undergoes latency, although the specific cells where latency occurs may vary [68]. Alphaherpesviruses are neurotropic and can infect both the central and peripheral nervous systems, enabling viral spread within the nervous system through the retrograde or antegrade transport of virions [68]. Viral glycoproteins serve the purpose of virus attachment and subsequent entry into host cells [69]. In the case of HSV-1, specific viral glycoproteins, including glycoprotein C (gC), glycoprotein B (gB), and glycoprotein D (gD), interact with host-cell entry receptors like herpesvirus entry mediator (HVEM) and nectin-1. Subsequently, viral glycoproteins gB, gH, and gL facilitate fusion, releasing virus particles into the cytoplasm. The HSV viral genome is subsequently transported into the nucleus, where replication commences to produce progeny virions [69].

### 3.1. Enhancing Safety and Tumor-Selective Replication of oHSV

Similar to other large DNA viruses, HSV-1 encodes multiple genes related to virulence and cellular tropism. To create safe oncolytic herpes simplex viruses (oHSVs), single or multiple genes are deleted or mutated to enhance safety, reduce virulence, and maintain selective replication and spread in tumor cells. The most commonly modified non-essential genes in oHSV, encoding infected-cell proteins ICP, are Herpesvirus RL1 (γ34.5), UL39, and α47, corresponding to ICP34.5, ICP6, and ICP47, respectively. Wild-type HSV uses these genes for host immune evasion [66]. ICP34.5 is a multifunctional protein that significantly contributes to HSV-1-associated neuro-virulence. Consequently, viruses lacking functional ICP34.5 are safer because they exhibit reduced infection and replication in the central nervous system [70]. Most oHSVs are constructed by the deletion or expression of ICP34.5 under tumor-specific promoters. The deletion of ICP34.5 leads to the development of a safe virus, albeit with significant attenuation in replication. ICP34.5 is involved in inhibiting the global shut-off of host protein synthesis in response to viral attacks. Wild-type HSV triggers the activation of PKR, which subsequently inactivates eIF-2α, resulting in the cessation of protein synthesis [71,72]. In the presence of ICP34.5, viral protein synthesis is restored as a result of the lack of eIF-2α phosphorylation. Deleting both copies of ICP34.5 hinders HSV-1’s ability to propagate in normal cells [66]. The defective virus can selectively replicate in tumor cells because of the flawed antiviral IFN response [53]. In Phase I clinical trials in glioma and melanoma patients, the virus lacking ICP34.5 in its genome has demonstrated safety [53].

UL39 gene-encoded protein ICP6 is the large sub-unit in viral ribonuclease reductase, which converts ribonucleotides into deoxynucleotides that are utilized in viral genome synthesis [73]. Tumor cells upregulate the expression of the cellular ribonuclease, resulting in tumor-selective replication of ICP6 mutant oHSVs [54]. ICP6 also inhibits TNF-mediated and Fas ligand-mediated apoptosis and necroptosis through its interaction with caspase 8 [54,73]. Thus, ICP6 is mutated in several oncolytic HSV candidates. ICP6-deleted oHSV is currently in Phase I clinical trials for liver metastases and primary liver cancers [54] (https://clinicaltrials.gov/ct2/show/NCT01071941 (accessed on 12 September 2023)). rRp450 is an oHSV recombinant with a deletion in ICP6 and insertion in rat cytochrome P450 2B1 (CYP2B1), a prodrug enzyme for cyclophosphamide (CPA). rRp450 has shown therapeutic efficacy in treating animal models of glioblastoma [74]. In a 2002 study by Pawlik et al., rRp450 was used to target colon carcinoma liver metastasis in combination with CPA. Treatment with rRp450 and CPA demonstrated therapeutic benefits in mouse models of diffuse colon carcinoma liver metastases [55].

HSV-1 downregulates MHC-I expression via ICP47. The ICP47 protein binds to the cellular transporter associated with antigen presentation (TAP), thereby blocking the antigenic peptides from being loaded onto nascent MHC class I molecules for presentation. In turn, the lack of MHC-I antigen presentation lessens the CD8+ T-cell immune response. Hence, the deletion of ICP47 enables efficient antigen loading, leading to enhanced CD8+ T-cell responses and antitumor immune reactions [56,75,76]. Furthermore, the deletion of ICP47 permits the early expression of US11, which can partially counteract cellular PKR-mediated antiviral responses, consequently offering partial restoration of ICP34.5 functioning in ICP34.5 mutant viruses [56]. G47∆ is a multimutated oncolytic HSV (oHSV) that carries a deletion for both copies of γ34.5 and α47 and inactivated ICP6. The ICP47 protein is a gene product of α47. G47∆ has demonstrated remarkable in vivo efficacy in inhibiting tumor growth using an immune-deficient mouse model of U87MG glioma. Additionally, G47∆ was effective in reducing the tumor burden in immunocompetent mouse models of Neuro2a neuroblastoma tumors [56].

### 3.2. Engineered oHSVs for Tumor-Selective Tropism

Viral entry of oHSV-1 is facilitated by virus-encoded glycoproteins gC, gD, gH/gL, and gB. These oHSV glycoproteins can be modified to modify viral tropism. HSV glycoproteins gD and gB bind to cell surface receptors such as HVEM and nectin-1 for viral entry. Nevertheless, HVEM and nectin-1 exhibit ubiquitous expression across most human tissues and cells. Therefore, the engineering of gD and gB facilitates tumor-specific binding and entry while safeguarding normal cells from infection. gD exhibits species-specificity and is widely considered the principal determinant of HSV tropism [77,78,79].

HER-2 is overexpressed in the majority of cancers, including glioblastoma, breast cancer, and ovarian cancer [80]. Two mutant oHSVs, oHSV-LM249 and oHSV-LM113, were engineered by introducing single-chain variable fragments (scFVs) targeting HER-2 into gD. In oHSV-LM113, a gD deletion is present in the N-terminal portion between amino acids 6-38 [80,81]. In the case of the oHSV-LM249 virus, a gD deletion is located in the core region between amino acids 61-218 [58]. These gD modifications enable selective infection of HER-2-positive tumors, even though the overall in vitro growth of recombinant viruses exhibited a one-log reduction compared to wild-type HSV [58,59]. Preclinical experiments have been conducted to showcase the impact of oHSV-LM113 and oHSV-LM249 on breast and ovarian cancers [57,58,59]. HSV gD has been additionally harnessed to target various other tumor-specific proteins, including EGFR (oHSV R611), CAE (oHSV-KNC), PSMA (oHSV-R593), and EGFRvIII (oHSV-KNE) [82]. Apart from gD, gB is involved in HSV binding, fusion, and entry and has also been engineered to target selective proteins in tumor cells [59].

Moreover, oHSVs have been created to express ICP34.5 under tumor-specific promoters, including promoters for Nestin and Musashi-1. This approach amplifies virus replication in cancer cells while sparing non-tumor cells [60]. rQNestin is another oHSV variant that retains ICP34.5 under Nestin promoter/enhancer element expression [83]. The variant has proven so far to be efficacious in the treatment of neurological tumors. Treatment with rQNestin34.5 significantly boosted the survival rate of nude mice to over 90%, as opposed to the 77.8% survival rate observed in mice carrying intracerebral human U87EGFR glioma [60].

## 4. Oncolytic Adenovirus

Rowe and colleagues first isolated adenovirus in 1953 while studying the growth of Polioviruses in the human adenoid tissues, during which a transmissible agent was found that caused a cytopathic effect in tissue in the absence of Poliovirus [84]. Adenoviruses are non-enveloped linear double-stranded DNA viruses with an average size of 70 nm. They contain an average genome size of 36 kbp, with viral transcription occurring in the host cell’s nucleus [84]. The adenovirus family is divided into seven species, including 57 human serotypes [85].

### 4.1. Molecular Basis for Adenovirus Cancer Tropism

Cellular receptors mediate the entry of adenovirus into human cells. Serotypes from species A, C, E, and F use Coxsackie–adenovirus receptors (CAR), while serotypes from B and D use receptors CD46/MCP, CD86, CD80, and DSG-2. All adenoviruses use integrin as their secondary receptor [86]. The first step of cell entry is the high-affinity attachment of the fiber protein to its primary receptor, CAR. This attachment allows the binding of cellular integrins such as αvβ3 and αvβ5 to the Arg-Gly-Asp (RGD) motif in the Ad penton base [87]. The interaction of Ad–integrin promotes internalization by endocytosis and the release of the virus into the cytoplasm, and, eventually, the genome is transported to the nucleus [87]. The CAR receptors are broadly expressed in the tight junction of epithelial tissues, and mutations to ablate the interaction between adenovirus and CAR receptors have been explored in the context of Adenovirus serotype 5 (Ad5). Ad5 mutant was developed to selectively target αvβ6 integrins expressed in metastatic pancreatic ductal adenocarcinoma (PDAC). The Ad5 mutant Ad5-3Δ-A20T infected and killed αvβ6 integrin expressing cells more than the wild-type Ad5 virus. Ad5-3Δ-A20T efficiently inhibited the growth of human pancreatic cancer xenograft in murine models [61].

### 4.2. Genetic Modifications for Cancer Tropism

Oncolytic adenoviruses are modified to infect and replicate in cancer cells selectively, but not in normal cells, and are named conditionally replicative adenoviruses (CRAds). The generation of CRAds is based on using tumor-specific promoters for replication or exploiting the interaction of essential viral genes with tumor-specific proteins. There are crucial viral genes expressed under tumor-specific promoters that will allow the replication of CRAds only in those cells. An example of such a promoter is cyclooxygenase-2 (Cox2) for gastric and pancreatic cancer [88,89]. Among other promoter elements are the hypoxia-inducible factor (HIF) responsive promoter in gliomas and medulloblastomas and human telomerase reverse transcriptase (hTERT) for bone- and soft-tissue sarcomas and other cancers [90]. The E1 protein critical for AdV replication has its native promoter replaced with the hTERT promoter. Inserting an hTERT promoter enhances AdV5 cancer tropism and tumor-specific replication [62]. AdV5 virus with the hTERT promoter driving the expression of E1A and E1B led to successful replication in a panel of human cancer cells but was significantly attenuated in normal human fibroblasts lacking telomerase activity. Intratumoral injection of AdV5 hTERT inhibited the growth of human lung tumor xenografts in mouse models [62].

Another approach involves deleting or mutating genes that are essential for productive AdV replication. For example, the E1 region consists of two genes, E1A and E1B, involved in replication. E1A-encoded proteins play a role in the transcription of early viral genes and stimulate infected cells to progress into the S phase. On the other hand, E1B-encoded proteins are associated with apoptosis through their interaction with cellular p53 and Rb proteins [91]. Both E1A and E1B encoded proteins work together for the successful replication of the virus. Mutant AdV5 lacking the E1 region has been used to target tumor cells with defective p53 or Rb genes. For instance, ONYX-15 lacks a 55kDa region of the E1B gene, enabling selective replication in tumors with mutated p53 [91]. ONYX-15 demonstrated antitumor effectiveness after intratumoral and intravenous administration in nude mouse–human tumor xenografts of cervical and laryngeal carcinoma where functional p53 was absent. ONYX-15 is currently in clinical trials for head and neck cancer [63].

Gendicine^®^ (rAd-p53) is a replication-incompetent adenovirus-based gene therapy drug that has received clinical approval for use in China. rAd-p53 expresses p53 using a Rous sarcoma virus promoter and its stimulating proteins instead of the virulent E1 gene within cancerous cells. The p53 pro-apoptotic boost helps in the elimination of the cells of interest. Gendicine^®^ is currently used in the treatment of various cancer types, including liver and head cancers [64].

## 5. Conclusions

While only a limited number of OVs have been approved for clinical use against specific tumor types, oncolytic virotherapy (OVT) remains underutilized, offering untapped potential. To further advance the development of OVT, several potential challenges must be addressed. These challenges include improving cellular tropism within the tumor environment, enhancing delivery to metastatic tumor sites, and activating potent antitumor immune responses. Understanding the cellular tropism of OVs is critical to overcoming these challenges and to further development of OVT. This is especially significant given the diverse cell populations within the tumor microenvironment, which include cancer stem cells, endothelial cells, fibroblasts, immune cells, extracellular matrices, and connective tissues. An ideal OV is expected to show robust replication in the tumor microenvironment (TME). Additionally, understanding the tropism of OVs is crucial because of the use of various carrier cells for OV delivery, including mesenchymal stem cells, neural stem cells, and chimeric antigen receptor T (CAR-T) cells. Gaining insights into the interaction between carrier cells and OVs could open up potential avenues for improved therapeutics.

Equally critical is comprehending how primary immune cells detect OVs and trigger antiviral immune responses, even before the viruses reach their intended tumor sites. A robust antiviral immune response can eliminate the virus and induce inflammation. Therefore, achieving a delicate balance between antiviral and antitumor immune responses is pivotal for enhancing the success of OVT. Comprehending the tropism of OVs for various cell types is essential to tackling these challenges.

In contrast to numerous other cancer therapies, OVs possess the potential to combat a wide range of cancer malignancies and can be effectively integrated with existing treatments, including immunotherapy and cell therapy. The integration of OVT with other therapies has the potential to overcome the limitations associated with those treatments. Hence, comprehending the tropism of OVs within the complex TME is indispensable for the successful development of combinational cancer therapies.

## Figures and Tables

**Figure 1 viruses-15-02262-f001:**
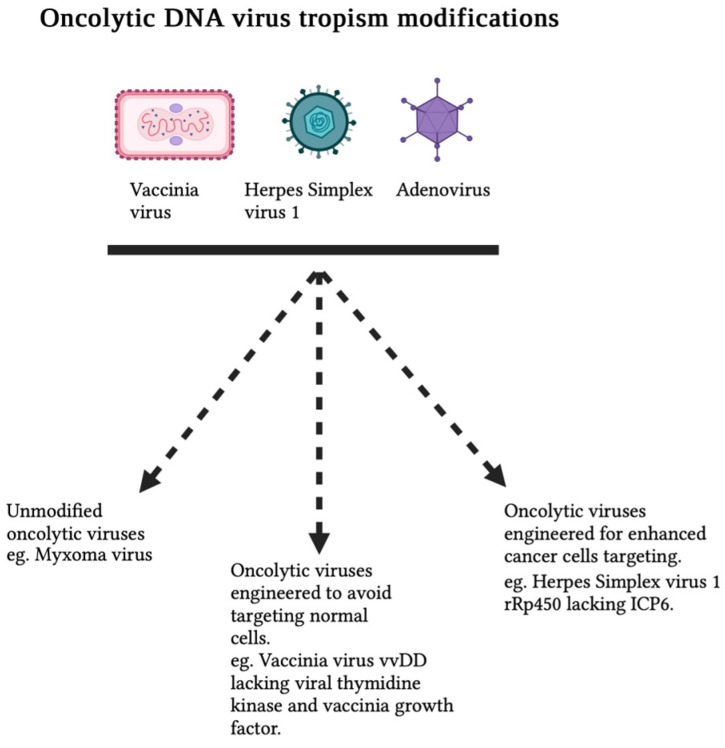
Oncolytic DNA virus cellular tropism. Oncolytic DNA viruses are engineered to selectively infect, replicate within, and eliminate cancer cells while sparing normal cells. Notable examples include vvDD (vaccinia virus double-deleted) and HSV-1 lacking ICP6 (infected-cell protein 6).

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
