# Peer review of "Tumor Tropism of DNA Viruses for Oncolytic Virotherapy"

_viruses, 2023, doi:10.3390/v15112262_

Round 1
Reviewer 1 Report
Comments and Suggestions for Authors
This manuscript offers an extensive literature review on the utilization of DNA viruses in oncolytic virotherapy development. It compiles and elaborates on numerous virus modifications aimed at enhancing the selectivity of virus tropism toward cancer cells through diverse strategies. The text well represents the many scientific publications in the field, it is logical and well structured. However, both the text and the illustration of the review could be improved. Having an illustration in a review is always a winning strategy. I liked the idea of presenting three different viral families with three representatives and their schematic images. However, the arrow from the center leading to the different variants of genetic modifications confuses the reader. It gives the impression that all the examples refer to a central virus located in the center: Herpes Simplex virus 1. However, this is a false sense, two of the viral examples Myxoma virus and Vaccinia belong to the poxviridae family, representatives of which on the left. I would like the authors to revise the figure so that the viral schemes in the top row and the examples of oncolytic viruses or their modified variants at the bottom coincide. Another option is to separate the top and bottom levels of the figure with a horizontal line. Do this in a way that does not give the impression that all examples relate only to the central viral image. I would also like to correct the title of the illustration. More appropriate title: “Oncolytic DNA virus tropism modifications” or “Examples of oncolytic DNA virus tropism modifications”.
Several examples of virus classifications are located throughout the text of the review. It would be a good idea to illustrate such textual classifications with diagrams. In this case, illustrations improve and speed up the reader's perception of the information.
In the attached file - 39 comments to the text are given directly in pdf format and these comments are the most essential. The following section on the quality of the English language suggests minor changes to the text.

Overall, the text is well written. However, in some places it is better to break large paragraphs into smaller ones, following the golden rule of one thought-one paragraph. There are also some incorrect formulations and inaccuracies in the text. In the attached text below, small modifications of some sentences of the review are offered, which speed up the reading and perception of the information of the review text. These modifications are not as important as the comments in the review text in pdf file itself. They frequently contain synonyms of words, and this is not so essential, original words might be even better. However, more important is the improved and corrected grammatical structure of the sentences, which allows faster reading comprehension. Below are 119 sentence improvements suggested.
Figure 1. A suggestion for the corrected title is in the pdf file.
Text inside the figure
Original: Oncolytic virus engineered to avoid normal cells
Suggested: Oncolytic virus engineered to avoid targeting normal cells
Text of the legend:
Original: Oncolytic DNA viruses are specifically designed to either prevent infection and replication in normal cells or to selectively infect, replicate within, and eliminate cancer cells. Some notable examples include vvDD (vaccinia virus double-deleted) and ICP6 (infected cell protein 6).
Suggested improvement: Oncolytic DNA viruses are engineered to selectively infect, replicate within, and eliminate cancer cells while sparing normal cells. Notable examples include vvDD (vaccinia virus double-deleted) and ICP6 (infected cell protein 6).
Line 22:
Original: "Due to their multi-mechanistic antitumor effects, oncolytic viruses (OVs) have recently emerged as novel cancer immunotherapy agents. OVs are from diverse families of viruses that selectively infect and kill cancer cells while sparing normal ones."
Suggested Improvement: "Owing to their multifaceted antitumor effects, oncolytic viruses (OVs) have recently gained prominence as innovative agents for cancer immunotherapy. These viruses belong to various families, selectively targeting and destroying cancer cells while leaving normal cells unharmed."
Line 28:
Original: "OVs are considered one of the emerging cancer treatment modalities that can be combined with most existing cancer treatment modalities, such as radiation, chemotherapy, surgery, immunotherapy, and cell-based therapies."
Suggested Improvement: " OVs are recognized as one of the emerging cancer treatment modalities that can complement most existing cancer therapies, including radiation, chemotherapy, surgery, immunotherapy, and cell-based treatments."
Line 34:
Original: "Above all, the critical attribute of all OVs is tumor-selective infection and replication."
Suggested Improvement: "Foremost among all characteristics, the pivotal feature of every OV is its ability to selectively infect and replicate within tumors."
Line 35:
Original: "This cancer tropism (oncotropism) of OVs has been achieved in multiple ways: i) engineering OVs to target cancer cells, either at the entry-level or during post-entry replication steps; ii) use of interferon (IFN) sensitive viruses that can’t replicate in normal cells but can replicate in cancer cells defective for those pathways; iii) engineering OVs lacking single or multiple immune evasion genes; iv) exploiting the metabolic pathways in cancer cells."
Suggested Improvement: "This cancer tropism, often referred to as oncotropism, in OVs has been accomplished through various approaches: i) engineering OVs to specifically target cancer cells, either at the entry stage or during post-entry replication steps; ii) utilizing IFN-sensitive viruses that cannot replicate in normal cells but thrive in cancer cells with defective IFN pathways; iii) crafting OVs devoid of one or more immune evasion genes; iv) harnessing the unique metabolic pathways of cancer cells."
Line 41:
Original: "Amongst the DNA viruses that have shown potential for cancer therapy are adenoviruses, parvoviruses, members of poxviruses such as vaccinia virus (VACV) and myxoma virus (MYXV), and members of herpesviruses. Large DNA genomes of poxviruses and herpesviruses allowed not only enhanced tropism for cancer cells but also severely reduced ability to infect normal cells, ultimately enhancing the safety of these oncolytic viruses."
Suggested Improvement: "DNA viruses that have demonstrated promise in cancer therapy include adenoviruses, parvoviruses, various poxviruses like vaccinia virus (VACV) and myxoma virus (MYXV), as well as herpesviruses. The large DNA genomes of poxviruses and herpesviruses not only enable enhanced tropism for cancer cells but also significantly reduce their ability to infect normal cells, thereby bolstering the safety of these oncolytic viruses."
Line 54:
Original: "2.1. Poxvirus natural tropism for cancer cells"
Suggested Improvement: "2.1. Natural Tropism of Poxviruses for Cancer Cells"
Line 68:
Original: "Oncolytic poxviruses can be broadly classified as replicating poxviruses, non-replicating vaccine vectors, and chimeric poxviruses."
Suggested Improvement: "Oncolytic poxviruses can be broadly categorized into replicating poxviruses, non-replicating vaccine vectors, and chimeric poxviruses."
Line 96:
Original: "Interestingly, both MYXV and VACV cannot bind primary human CD34+ stem cells."
Suggested Improvement: "Notably, neither MYXV nor VACV can bind to primary human CD34+ stem cells."
Line 100:
Original: "Poxvirus binding, entry, infection, and replication is a multistep process of attachment, internalization, and virion core release into the cytoplasm. At least four virus-encoded proteins, D8, A27, H3, and A26, are involved in poxvirus cell attachment."
Suggested Improvement: "The process of poxvirus binding, entry, infection, and replication is a multistep sequence involving attachment, internalization, and release of virion cores into the cytoplasm. This process relies on at least four virus-encoded proteins: D8, A27, H3, and A26, which play pivotal roles in poxvirus cell attachment."
Line 109:
Original: "These results suggest that for poxviruses, infection of normal primary cells is controlled at multiple levels of virus binding, entry, replication, and innate immune signaling pathways, many of which cancer cells are missing."
Suggested Improvement: "These findings imply that poxviruses exert control over the infection of normal primary cells at various stages, including virus binding, entry, replication, and innate immune signaling pathways, many of which are deficient in cancer cells."
Line 110:
Original: "Vaccinia has been the subject of most studies on poxvirus biology and pathogenesis, primarily because of the use of VACV as a vaccine against smallpox."
Suggested Improvement: "Vaccinia has been the primary focus of most studies on poxvirus biology and pathogenesis, largely due to its use as a smallpox vaccine."
Line 118:
Original: "Among all these different forms, single-layered intracellular IMVs are the most abundant forms and are released after cell lysis."
Suggested Improvement: "Of all these various forms, the single-layered intracellular IMVs are the most abundant and are released following cell lysis."
Line 119:
Original: "The released IMVs can efficiently infect neighboring cells via endocytosis."
Suggested Improvement: "These released IMVs can efficiently infect neighboring cells through the process of endocytosis."
Line 120:
Original: "A small fraction of IMVs is wrapped with the double-layer Golgi membrane to form IEVs, which are transported to the cell periphery to become CEVs."
Suggested Improvement: "A minority of IMVs is enveloped by the double-layer Golgi membrane, forming IEVs, which are then transported to the cell periphery to become CEVs."
Line 122:
Original: "Another smaller fraction of IMVs directly move to the cell periphery, fuse to the cell membrane, acquire an outer envelope, and are released as EEVs."
Suggested Improvement: "A smaller fraction of IMVs directly moves to the cell periphery, fuses with the cell membrane, acquires an outer envelope, and is released as EEVs."
Line 123:
Original: "The EEV of VACV has evolved for rapid systemic spread within the host and evasion of immune-mediated clearance."
Suggested Improvement: "VACV's EEV has evolved to facilitate rapid systemic spread within the host and to evade immune-mediated clearance."
Line 126:
Original: "VACV EEV is resistant to complement activation as it incorporates host regulators of complement activation (RCAs) proteins such as CD46, CD55, CD59, CD71, CD81, and MCH I into its envelope."
Suggested Improvement: "VACV EEV is resistant to complement activation due to the incorporation of host regulators of complement activation (RCAs) proteins, including CD46, CD55, CD59, CD71, CD81, and MCH I, into its envelope."
Line 128:
Original: "To further enhance the spread of VACV EEV, several approaches have been used to enhance virus spread in the tumor bed."
Suggested Improvement: To further improve oncolytic properties of VACV EEV, several approaches have been used to enhance virus spread in the tumor bed."
Line 135:
Original: "Previous studies demonstrated that deletion of B18R and expression of IFN-β resulted in a virus that cannot infect normal cells but replicates in cancer cells and enhanced tumor killing."
Suggested Improvement: "Previous studies demonstrated that deleting B18R and expressing IFN-β resulted in a virus that cannot infect normal cells but can replicate within cancer cells, enhancing tumor killing."
Line 136:
Original: "Another EEV-associated protein, B5R, responsible for viral morphogenesis, trafficking, and dissemination and involved in the non-specific targeting of healthy and tumor cells, was engineered for tumor-specific replication, escaping virus neutralization but retaining oncolytic activity."
Suggested Improvement: "Another EEV-associated protein, B5R, responsible for viral morphogenesis, trafficking, and dissemination, and involved in the non-specific targeting of healthy and tumor cells, was engineered to enable tumor-specific replication, evading virus neutralization while retaining oncolytic activity."
Line 142:
Original: "As described before, poxviruses, notably different strains of VACVs, can inherently infect tumor cells."
Suggested Improvement: "As mentioned previously, poxviruses, particularly various strains of VACVs, naturally possess the ability to infect tumor cells."
Line 144:
Original: "An attenuated double-deleted VACV WR strain (vvDD) was generated for selectively targeting and killing cancer cells."
Suggested Improvement: "An attenuated VACV WR strain with two deletions (vvDD) was developed to selectively target and eliminate cancer cells."
Line 146:
Original: "Deletion of TK led to significant attenuation in normal, non-dividing cells but retained replication ability in tumor cells, which are dividing cells."
Suggested Improvement: "Deleting TK resulted in significant attenuation in normal, non-dividing cells, while retaining replication ability in tumor cells, which are actively dividing."
Line 150:
Original: "Overall, the deletion of both TK and VGF created an oncolytic VACV that is highly tumor-selective."
Suggested Improvement: "In summary, the deletion of both TK and VGF produced an oncolytic VACV with a high degree of tumor selectivity."
Line 160:
Original: "Apart from VACV, TK was deleted from other orthopoxviruses such as CPXV and raccoon poxvirus for tumor-selective replication."
Suggested Improvement: "Besides VACV, TK was also removed from other orthopoxviruses like CPXV and raccoon poxvirus to achieve tumor-selective replication."
Line 161:
Original: "To further attenuate VACV in normal cells but retain replication in tumor cells, both VGF and O1 genes were deleted."
Suggested Improvement: "To further weaken VACV's activity in normal cells while maintaining replication in tumor cells, both VGF and O1 genes were removed."
Line 162:
Original: "VGF functions in a paracrine and autocrine to enhance virus replication by activating the EGFR-dependent mitogen-activated protein kinase (MAPK)-ERK pathway."
Suggested Improvement: "VGF acts in a paracrine and autocrine manner to enhance virus replication by activating the EGFR-dependent mitogen-activated protein kinase (MAPK)-ERK pathway."
Line 164:
Original: "The O1 protein is required for sustained activation of extracellular signal-regulated kinase 1/2 (ERK1/2) signaling initiated by VGF."
Suggested Improvement: "The O1 protein is essential for sustaining the activation of extracellular signal-regulated kinase 1/2 (ERK1/2) signaling initiated by VGF."
Line 167:
Original: "Thus, deletion of both VGF and O1 allowed MAPK-dependent tumor-selective replication of VACV, but not in normal cells."
Suggested Improvement: "Hence, the deletion of both VGF and O1 enabled MAPK-dependent tumor-specific replication of VACV, while sparing normal cells."
Line 169:
Original: "The efficacy of this virus was tested in a pancreatic ductal adenocarcinoma xenograft mouse model."
Suggested Improvement: "The effectiveness of this virus was assessed in a xenograft mouse model of pancreatic ductal adenocarcinoma."
Line 171:
Original: "Besides killing cancer cells, this oncolytic VACV can infect and destroy tumor vascular endothelial cells."
Suggested Improvement: "In addition to eradicating cancer cells, this oncolytic VACV can also infect and eliminate tumor vascular endothelial cells."
Line 175:
Original: "Furthermore, another modified VACV that had deletion of four viral genes (A48R, B18R, C11R, and J2R) involved in different pathways such as metabolic, proliferation, and signaling showed antitumor activity and at the same time retained tumor selectivity in vivo."
Suggested Improvement: "Moreover, another modified VACV, with the deletion of four viral genes (A48R, B18R, C11R, and J2R) involved in various pathways like metabolism, proliferation, and signaling, demonstrated antitumor activity while maintaining tumor selectivity in vivo."
Line 180:
Original: "A recent study demonstrated that the deletion of three key immune evasion gene products (C10L, N2L, and C6L) from VACV retained replication ability in cancer cells."
Suggested Improvement: "A recent study demonstrated that deleting three key immune evasion gene products (C10L, N2L, and C6L) from VACV preserved its replication ability in cancer cells."
Line 182:
Original: "VACV-encoded proteins C10, A46, N2L, and C6 antagonized the TLR3-IRF3 signaling pathway at different levels."
Suggested Improvement: "Proteins encoded by VACV, including C10, A46, N2L, and C6, acted as antagonists in the TLR3-IRF3 signaling pathway at various stages."
Line 183:
Original: "Deleting these genes allowed phosphorylation of IRF3, the key protein involved in the activation of TLR3, resulting in enhanced cytotoxic T lymphocyte (CTL) responses in a syngeneic mouse model."
Suggested Improvement: "The deletion of these genes facilitated the phosphorylation of IRF3, a pivotal protein in the activation of TLR3, leading to heightened cytotoxic T lymphocyte (CTL) responses in a syngeneic mouse model."
Line 185:
Original: "Another study took a similar approach to delete multiple immunomodulatory genes (N1L, K1L, K3L, A46R, and A52R)."
Suggested Improvement: "Another study adopted a similar approach by deleting multiple immunomodulatory genes (N1L, K1L, K3L, A46R, and A52R)."
Line 191:
Original: "Deleting immune evasion genes can activate antitumor immune responses through virus-induced cytokine production."
Suggested Improvement: "The deletion of immune evasion genes can activate antitumor immune responses through virus-induced cytokine production."
Line 192:
Original: "For example, the oncolytic VACV construct deleted N1L protein expression (VVΔTKΔN1L) showed an enhanced virus-induced cytokine production, which upregulated the amount of circulating NK cells in surgery-induced metastatic models of cancer."
Suggested Improvement: "As an example, the oncolytic VACV construct with deleted N1L protein expression (VVΔTKΔN1L) exhibited enhanced virus-induced cytokine production, resulting in an increase in the number of circulating NK cells in surgery-induced metastatic models of cancer."
Line 195:
Original: "Furthermore, the expression of the cytokine IL-12 using this N1LKO background (VVΔTKΔN1L-IL12) significantly improved efficacy in the head and neck cancer model."
Suggested Improvement: "Moreover, the expression of the cytokine IL-12 within this N1LKO background (VVΔTKΔN1L-IL12) significantly improved efficacy in the head and neck cancer model."
Line 197:
Original: "VACV N1L is important in immune evasion and is essential for poxvirus virulence."
Suggested Improvement: "VACV N1L plays a crucial role in immune evasion and is indispensable for poxvirus virulence."
Line 200:
Original: "Deletion of immunomodulatory genes can enhance tumor-specific tropism and oncolytic activity of poxviruses."
Suggested Improvement: "The deletion of immunomodulatory genes can enhance the tumor-specific tropism and oncolytic activity of poxviruses."
Line 203:
Original: "Members of Yatapoxviruses, such as Tanapoxvirus (TPV) and Yabalike disease virus (YLDV), have been exploited for oncolytic virotherapy."
Suggested Improvement: "Yatapoxviruses, including Tanapoxvirus (TPV) and Yabalike disease virus (YLDV), have been harnessed for oncolytic virotherapy."
Line 205:
Original: "These viruses cause a relatively benign infection in humans but infect and kill cancer cells."
Suggested Improvement: "While these viruses typically result in a relatively benign human infection, they can infect and eliminate cancer cells."
Line 206:
Original: "TPV cellular tropism was tested with primary human dermal fibroblasts (pHDFs) and peripheral blood mononuclear cells (PBMCs)."
Suggested Improvement: "The cellular tropism of TPV was examined using primary human dermal fibroblasts (pHDFs) and peripheral blood mononuclear cells (PBMCs)."
Line 207:
Original: "pHDFs were permissive for TPV infection and replication."
Suggested Improvement: "pHDFs supported TPV infection and replication."
Line 208:
Original: "When PBMCs were infected with TPV, monocytes (CD14+) were the major population infected with TPV."
Suggested Improvement: "Upon TPV infection of PBMCs, monocytes (CD14+) constituted the predominant infected population."
Line 209:
Original: "However, the virus failed to replicate when monocytes were differentiated into macrophages."
Suggested Improvement: "Nevertheless, the virus did not replicate when monocytes were differentiated into macrophages."
Line 211:
Original: "In vitro, TPV can replicate in multiple human cancer cell lines."
Suggested Improvement: "In vitro, TPV can replicate within various human cancer cell lines."
Line 215:
Original: "TPV-encoded neuregulin (NRG), an EGF-like growth factor (encoded by 15L), which enhances various types of cell proliferation, has been deleted (TPVΔ15L), allowing enhanced melanoma tumor regression than wild-type TPV."
Suggested Improvement: "The deletion of TPV-encoded neuregulin (NRG), an EGF-like growth factor (encoded by 15L) that enhances various types of cell proliferation, as seen in TPVΔ15L, resulted in enhanced regression of melanoma tumors compared to wild-type TPV."
Line 217:
Original: "Deletion of viral type I IFN binding receptor (136R) can also make selective replication in tumors and an abortive replication in primary cells that respond to IFN."
Suggested Improvement: "Additionally, the deletion of the viral type I IFN binding receptor (136R) enables selective replication in tumors and abortive replication in primary cells responsive to IFN."
Line 218:
Original: "Unlike TPV, the oncolytic potential of YLDV has been tested in a limited number of studies."
Suggested Improvement: "In contrast to TPV, the oncolytic potential of YLDV has been evaluated in a limited number of studies."
Line 221:
Original: "A screening using Parapoxvirus ovis (strain D1701) demonstrated antitumor activity in human xenograft models and syngeneic B16-F10 melanoma models."
Suggested Improvement: "Screening using Parapoxvirus ovis (strain D1701) revealed antitumor activity in human xenograft models and syngeneic B16-F10 melanoma models."
Line 224:
Original: "Oncolytic MYXV, a member of Leporipoxvirus, has a natural tropism for cancer cells and has been tested for oncolytic activity in several preclinical cancer models."
Suggested Improvement: "Oncolytic MYXV, a member of Leporipoxvirus, exhibits a natural affinity for cancer cells and has undergone testing for oncolytic activity in various preclinical cancer models."
Line 237:
Original: "Another chimeric orthopoxvirus was generated by coinfecting CV-1 cells with multiple (nine) strains of orthopoxviruses, namely cowpox virus strain Brighton, raccoon poxvirus strain Herman, rabbitpox virus strain Utrecht, VACV virus strains Western Reserve (WR), International Health Department (IHD), Elstree, CL, Lederle-Chorioallantoic (LC) and AS."
Suggested Improvement: "A chimeric orthopoxvirus was created by co-infecting CV-1 cells with multiple (nine) orthopoxvirus strains, including cowpox virus strain Brighton, raccoon poxvirus strain Herman, rabbitpox virus strain Utrecht, and various VACV strains: Western Reserve (WR), International Health Department (IHD), Elstree, CL, Lederle-Chorioallantoic (LC), and AS."
Line 241:
Original: "From more than 100 isolates CF33 and CF17 were selected based on the superior killing of the NCI-60 panel of cancer cell lines."
Suggested Improvement: "Out of over 100 isolates, CF33 and CF17 were chosen due to their exceptional ability to kill cancer cell lines from the NCI-60 panel."
Line 242:
Original: "The isolates were also tested against human pancreatic cancer and TNBC cell lines."
Suggested Improvement: "These isolates were also assessed against human pancreatic cancer and TNBC cell lines."
Line 244:
Original: "Similarly, CF17 also showed enhanced cell killing and efficacy in the IP ovarian cancer model."
Suggested Improvement: "Likewise, CF17 exhibited improved cell-killing abilities and effectiveness in the intraperitoneal (IP) ovarian cancer model."
Line 246:
Original: "Sequencing of the genome of CF33 revealed that the virus genome is derived mostly from three strains of VACV: IHD, WR, and Lister. No sequence was detected from the Raccoon poxvirus genome."
Suggested Improvement: "Genome sequencing of CF33 indicated that the virus genome is primarily derived from three VACV strains: IHD, WR, and Lister, with no sequences detected from the Raccoon poxvirus genome."
Line 248:
Original: "A similar approach was used to generate chimeric poxvirus deVV5 by recombining four VACV strains: WR, Wyeth, MVA, and Copenhagen."
Suggested Improvement: "A similar method was employed to create the chimeric poxvirus deVV5, achieved through recombination of four VACV strains: WR, Wyeth, MVA, and Copenhagen."
Line 250:
Original: "Furthermore, TK was deleted from deVV5 and armed with a suicide gene FCU1 (VV5-fcu1), which resulted in attenuation in normal cells."
Suggested Improvement: "Moreover, the deletion of TK from deVV5 and the incorporation of a suicide gene FCU1 (resulting in VV5-fcu1) led to attenuation in normal cells."
Line 252:
Original: "However, even after making these chimeric viruses and selecting recombinant virus isolates with the highest replication ability and cytotoxicity to cancer cells, these viruses successfully replicate and kill only a small number of available cancer cell lines against a particular type of cancer."
Suggested Improvement: "Nevertheless, despite the creation of these chimeric viruses and the selection of recombinant virus isolates with the highest replication capability and cytotoxicity against cancer cells, these viruses effectively replicate and eliminate only a limited subset of available cancer cell lines specific to a particular cancer type."
Line 255:
Original: "These findings suggest that rather than creating global changes in the viral genome, knockout or knock-in of selected viral genes to enhance cancer cell tropism and cytotoxicity is still a helpful strategy."
Suggested Improvement: "These findings indicate that, instead of implementing extensive alterations in the viral genome, the knockout or insertion of specific viral genes to enhance cancer cell tropism and cytotoxicity remains a valuable strategy."
Line 260:
Original: "The mammalian herpesviruses are members of the Herpesviridae family of viruses, classified into three subfamilies: alphaherpesviruses, betaherpesviruses, and gammaherpesviruses."
Suggested Improvement: "Mammalian herpesviruses belong to the Herpesviridae family of viruses and are categorized into three subfamilies: alphaherpesviruses, betaherpesviruses, and gammaherpesviruses."
Line 263:
Original: "Members of human alphaherpesviruses include HSV-1, HSV-2, and varicella-zoster virus (VZV)."
Suggested Improvement: "Human alphaherpesviruses comprise HSV-1, HSV-2, and varicella-zoster virus (VZV)."
Line 264:
Original: "HSV-1 has been established as an OV after successful clinical trials against multiple cancer types. The two oncolytic HSV-1-derived viruses approved for cancer treatment are T-VEC (Talimogene Laherparepvec) and G47Δ."
Suggested Improvement: "HSV-1 has gained recognition as an oncolytic virus (OV) following successful clinical trials against various cancer types. Two oncolytic HSV-1-derived viruses approved for cancer treatment are T-VEC (Talimogene Laherparepvec) and G47Δ."
Line 265:
Original: "Herpesviruses are enveloped virions with a large dsDNA genome of about 152 kb."
Suggested Improvement: "Herpesviruses are enveloped virions with a substantial dsDNA genome of approximately 152 kb."
Line 267:
Original: "Unlike poxviruses, they replicate in the nucleus of the infected cells."
Suggested Improvement: "In contrast to poxviruses, herpesviruses replicate within the nucleus of infected cells."
Line 268:
Original: "All members of the Herpesviridae establish latency, but the cells in which they establish latency vary."
Suggested Improvement: "Every member of the Herpesviridae family undergoes latency, although the specific cells where latency occurs may vary."
Line 269:
Original: "Members of alphaherpesviruses are neurotropic and can infect both the central and peripheral nervous systems, allowing viral spread within the nervous system through retrograde or antegrade transport of virions."
Suggested Improvement: "Alphaherpesviruses are neurotropic and can infect both the central and peripheral nervous systems, enabling viral spread within the nervous system through retrograde or antegrade transport of virions."
Line 271:
Original: "Viral glycoproteins are used for virus attachment and later entry into the cell."
Suggested Improvement: "Viral glycoproteins serve the purpose of virus attachment and subsequent entry into host cells."
Line 272:
Original: "Viral glycoproteins on HSV-1, glycoprotein C (gC), glycoprotein B (gB), and glycoprotein D (gD) bind to the host cells entry receptors such as herpesvirus entry mediator (HVEM) and Nectin-1."
Suggested Improvement: "In the case of HSV-1, specific viral glycoproteins, including glycoprotein C (gC), glycoprotein B (gB), and glycoprotein D (gD), interact with host cell entry receptors like herpesvirus entry mediator (HVEM) and Nectin-1."
Line 274:
Original: "Next, the viral glycoproteins gB, gH, and gL help in fusion releasing virus particles into the cytoplasm."
Suggested Improvement: "Subsequently, viral glycoproteins gB, gH, and gL facilitate fusion, releasing virus particles into the cytoplasm."
Line 276:
Original: "The HSV viral genome is then translocated into the nucleus, where it begins replication for making progeny virions."
Suggested Improvement: "The HSV viral genome is subsequently transported into the nucleus, where replication commences to produce progeny virions."
Line 279:
Original: "Like other large DNA viruses, HSV-1 encodes multiple genes associated with virulence and cellular tropism. To develop oHSVs as a safe oncolytic virus, single or multiple genes are deleted or mutated to increase safety and reduce virulence while retaining selective replication and spread in tumor cells."
Suggested Improvement: "Similar to other large DNA viruses, HSV-1 encodes multiple genes related to virulence and cellular tropism. To create safe oncolytic herpes simplex viruses (oHSVs), single or multiple genes are deleted or mutated to enhance safety, reduce virulence, and maintain selective replication and spread in tumor cells."
Line 281:
Original: "Herpesvirus RL1 (34.5), UL39, and 47 are the most frequently modified non-essential genes in oHSV encoding infected-cell proteins ICP: ICP34.5, ICP6, and ICP47, respectively."
Suggested Improvement: "The most commonly modified non-essential genes in oHSV, encoding infected-cell proteins ICP, are Herpesvirus RL1 (γ34.5), UL39, and α47, corresponding to ICP34.5, ICP6, and ICP47, respectively."
Line 285:
Original: "The ICP34.5 is a multifunctional protein that plays a significant role in HSV-1-associated neuro-virulence. Thus, viruses lacking functional ICP34.5 are safer due to reduced infection and replication in the central nervous system."
Suggested Improvement: "ICP34.5 is a multifunctional protein that significantly contributes to HSV-1-associated neuro-virulence. Consequently, viruses lacking functional ICP34.5 are safer because they exhibit reduced infection and replication in the central nervous system."
Line 290:
Original: "Deletion of ICP34.5 results in a safe virus but is highly attenuated in replication."
Suggested Improvement: "Deletion of ICP34.5 leads to the development of a safe virus, albeit with significant attenuation in replication."
Line 292:
Original: "The wild-type HSV causes the activation of PKR, which inactivates eIF-2 leading to a shutdown of protein synthesis."
Suggested Improvement: "Wild-type HSV triggers the activation of PKR, which subsequently inactivates eIF-2α, resulting in the cessation of protein synthesis."
Line 300:
Original: "ICP6 also blocks TNF-mediated and Fas ligand-mediated apoptosis and necroptosis by interaction with caspase 8."
Suggested Improvement: "ICP6 also inhibits TNF-mediated and Fas ligand-mediated apoptosis and necroptosis through its interaction with caspase 8."
Line 309:
Original: "Treatment with rRp450 and CPA showed therapeutic benefit in mouse models of diffuse colon carcinoma liver metastases."
Suggested Improvement: "Treatment with rRp450 and CPA demonstrated therapeutic benefits in mouse models of diffuse colon carcinoma liver metastases."
Line 315:
Original: "Thus, the deletion of ICP47 allows efficient antigen loading for better CD8+ T cell and antitumor immune responses."
Suggested Improvement: "Hence, the deletion of ICP47 enables efficient antigen loading, leading to enhanced CD8+ T cell responses and antitumor immune reactions."
Line 316:
Original: "Deletion of ICP47 also allows early expression of US11, which can then partially antagonize cellular PKR-mediated anti-viral responses and thus partially rescue the ICP34.5 function in the ICP34.5 mutant viruses."
Suggested Improvement: "Furthermore, the deletion of ICP47 permits early expression of US11, which can partially counteract cellular PKR-mediated antiviral responses, consequently offering partial restoration of ICP34.5 function in ICP34.5 mutant viruses."
Line 322:
Original: "G47 has shown high efficacy in vivo at inhibiting tumor growth in an immune-deficient mouse model of U87MG glioma tumor."
Suggested Improvement: "G47 has demonstrated remarkable in vivo efficacy in inhibiting tumor growth using an immune-deficient mouse model of U87MG glioma."
Line 323:
Original: "G47 was also effective at reducing the tumor burden in mice immunocompetent models of Neuro2a neuroblastoma tumors."
Suggested Improvement: "Additionally, G47 was effective in reducing the tumor burden in immunocompetent mouse models of Neuro2a neuroblastoma tumors."
Line 326:
Original: "The entry of oHSV-1 is mediated by virus-encoded glycoproteins gC, gD, gH/gL, and gB. oHSV glycoproteins can be engineered to alter the viral tropism."
Suggested Improvement: "Viral entry of oHSV-1 is facilitated by virus-encoded glycoproteins gC, gD, gH/gL, and gB. These oHSV glycoproteins can be modified to modify viral tropism."
Line 328:
Original: "However, HVEM and Nectin-1 are ubiquitously expressed in most human tissues and cells; thus, engineering gD and gB allows tumor-specific binding and entry and protects normal cells from infection."
Suggested Improvement: "Nevertheless, HVEM and Nectin-1 exhibit ubiquitous expression across most human tissues and cells. Therefore, the engineering of gD and gB facilitates tumor-specific binding and entry while safeguarding normal cells from infection."
Line 331:
Original: "gD is species-specific and is thought to be the primary determinant in HSV tropism."
Suggested Improvement: "gD exhibits species-specificity and is widely considered to be the principal determinant of HSV tropism."
Line 333:
Original: "HER-2 is over-expressed in most cancers, including glioblastoma, breast, and ovarian."
Suggested Improvement: "HER-2 is overexpressed in the majority of cancers, including glioblastoma, breast cancer, and ovarian cancer."
Line 335:
Original: "Two gD mutant oHSVs (oHSV-LM249 and oHSV-LM113) were constructed by inserting scFV targeting HER2 into gD. oHSV-LM113 contains a gD deletion between amino acids 6-38 of the N-terminal portion. The oHSV-LM249 virus contains a gD deletion in the core region between amino acids 61-218. These modifications of gD allow for selective infection of HER2-positive tumors, although the overall in vitro growth of recombinant viruses showed one log lower than wt-HSV."
Suggested Improvement: "Two mutant oHSVs, oHSV-LM249 and oHSV-LM113, were engineered by introducing single-chain variable fragments (scFV) targeting HER2 into gD. In oHSV-LM113, a gD deletion is present in the N-terminal portion between amino acids 6-38. In the case of the oHSV-LM249 virus, a gD deletion is located in the core region between amino acids 61-218. These gD modifications enable selective infection of HER2-positive tumors, even though the overall in vitro growth of recombinant viruses exhibited a one-log reduction compared to wild-type HSV."
Line 340:
Original: "Preclinical experiments have been presented that demonstrate the effect of oHSV-LM113 and oHSV-LM249 against breast and ovarian cancers."
Suggested Improvement: "Preclinical experiments have been conducted to showcase the impact of oHSV-LM113 and oHSV-LM249 on breast and ovarian cancers."
Line 341:
Original: "HSV gD has been further exploited to target other tumor-specific proteins such as EGFR (oHSV R611), CAE (oHSV-KNC), PSMA (oHSV-R593), EGFRvIII (oHSV-KNE)."
Suggested Improvement: "HSV gD has been additionally harnessed to target various other tumor-specific proteins, including EGFR (oHSV R611), CAE (oHSV-KNC), PSMA (oHSV-R593), and EGFRvIII (oHSV-KNE)."
Line 346:
Original: "Furthermore, oHSVs have been generated expressing ICP34.5 under tumor-specific promoters such as promoters for Nestin and Musashi-1 to increase virus replication in cancer cells but not in tumor cells."
Suggested Improvement: "Moreover, oHSVs have been created to express ICP34.5 under tumor-specific promoters, including promoters for Nestin and Musashi-1. This approach amplifies virus replication in cancer cells while sparing non-tumor cells."
Line 350:
Original: "rQNestin34.5 treatment increased the survival rate of nude mice by >90% of 77.8% of mice bearing intracerebral human U87EGFR glioma."
Suggested Improvement: "Treatment with rQNestin34.5 significantly boosted the survival rate of nude mice to over 90%, as opposed to the 77.8% survival rate observed in mice carrying intracerebral human U87EGFR glioma."
Line 390:
Original: "expression of E1A and E1B successfully replicated in a panel of human cancer cells but was highly attenuated in normal human fibroblast lacking telomerase activity."
Suggested Improvement: "The expression of E1A and E1B led to successful replication in a panel of human cancer cells but was significantly attenuated in normal human fibroblasts lacking telomerase activity."
Line 392:
Original: "AdV5 hTERT intratumoral injection inhibited human lung tumor xenograft growth in mouse models."
Suggested Improvement: "Intratumoral injection of AdV5 hTERT inhibited the growth of human lung tumor xenografts in mouse models."
Line 394:
Original: "Another approach is the deletion or mutations of genes essential for productive AdV replication."
Suggested Improvement: "Another approach involves deleting or mutating genes that are essential for productive AdV replication."
Line 396:
Original: "E1A-encoded proteins are involved in the transcription of early viral genes and stimulate infected cells to move to the S phase. E1B-encoded proteins are involved in apoptosis by interaction with cellular p53 and Rb proteins."
Suggested Improvement: "E1A-encoded proteins play a role in the transcription of early viral genes and stimulate infected cells to progress into the S phase. On the other hand, E1B-encoded proteins are associated with apoptosis through their interaction with cellular p53 and Rb proteins."
Line 398:
Original: "Both E1A and E1B encoded proteins function coordinately for successful virus replication."
Suggested Improvement: "Both E1A and E1B encoded proteins work together for the successful replication of the virus."
Line 399:
Original: "Tumor cells that are defective for the p53 or Rb gene were targeted with mutant AdV5 missing the E1 region."
Suggested Improvement: "Mutant AdV5 lacking the E1 region was used to target tumor cells with defective p53 or Rb genes."
Line 401:
Original: "For example, ONYX-15 lacks a 55kDa region of the E1B gene and can selectively replicate in p53 mutated tumors."
Suggested Improvement: "For instance, ONYX-15 lacks a 55kDa region of the E1B gene, enabling selective replication in tumors with mutated p53."
Line 402:
Original: "ONYX-15 showed antitumor efficacy following intratumoral and intravenous administration of nude mouse-human tumor xenografts of cervical and laryngeal carcinoma lacking functional p53."
Suggested Improvement: "ONYX-15 demonstrated antitumor effectiveness after intratumoral and intravenous administration in nude mouse-human tumor xenografts of cervical and laryngeal carcinoma, where functional p53 was absent."
Line 405:
Original: "Gendicine (rAd-p53) is a replication-incompetent Adenovirus gene therapy drug approved for clinical use in China."
Suggested Improvement: "Gendicine (rAd-p53) is a replication-incompetent adenovirus-based gene therapy drug that has received clinical approval for use in China."
Line 407:
Original: "rAd-p53 expresses p53 from a Rous sarcoma virus promoter and its stimulatory proteins in the place of the virulent E1 gene in cancerous cells."
Suggested Improvement: "rAd-p53 expresses p53 using a Rous sarcoma virus promoter and its stimulating proteins instead of the virulent E1 gene within cancerous cells."
Line 409:
Original: "Gendicine is currently used to treat many cancer types, including liver and head cancers."
Suggested Improvement: "Gendicine is currently used in the treatment of various cancer types, including liver and head cancers."
Line 412:
Original: "Although only a limited number of OVs are approved for clinical use against selected tumor types, oncolytic virotherapy (OVT) has yet to be exploited to its fullest potential."
Suggested Improvement: "While only a limited number of OVs have been approved for clinical use against specific tumor types, oncolytic virotherapy (OVT) remains underutilized, offering untapped potential."
Line 413:
Original: "For further development of OVT, several potential hurdles need to be overcome."
Suggested Improvement: "To further advance the development of OVT, several potential challenges must be addressed."
Line 414:
Original: "For example, enhancing cellular tropism in the tumor bed, enhancing delivery to the metastatic tumor sites, and activation of potent antitumor immune responses."
Suggested Improvement: "These challenges include improving cellular tropism within the tumor environment, enhancing delivery to metastatic tumor sites, and activating potent antitumor immune responses."
Line 418:
Original: "Particularly concerning the heterogeneous cell populations in the tumor microenvironment consisting of cancer stem cells, endothelial cells, fibroblasts, immune cells, extracellular matrices, and connective tissues."
Suggested Improvement: "This is especially significant given the diverse cell populations within the tumor microenvironment, which include cancer stem cells, endothelial cells, fibroblasts, immune cells, extracellular matrices, and connective tissues."
Line 421:
Original: "Another reason for studying and understanding the tropism of OVs is that different carrier cells are used for OV delivery, such as mesenchymal stem cells, neural stem cells, and chimeric antigen receptor T (CAR-T) cells."
Suggested Improvement: "Additionally, understanding the tropism of OVs is crucial due to the use of various carrier cells for OV delivery, including mesenchymal stem cells, neural stem cells, and chimeric antigen receptor T (CAR-T) cells."
Line 423:
Original: "Understanding how carrier cells interact with OV would provide potential avenues for better therapeutics."
Suggested Improvement: "Gaining insights into the interaction between carrier cells and OV could open up potential avenues for improved therapeutics."
Line 425:
Original: "It is also critical to understand how primary immune cells recognize OV and activate the anti-viral immune responses, even before the viruses reach the target tumor sites."
Suggested Improvement: "Equally critical is comprehending how primary immune cells detect OVs and trigger antiviral immune responses, even before the viruses reach their intended tumor sites."
Line 426:
Original: "A strong anti-viral immune response can clear the virus and cause inflammation. Thus, maintaining a balance between the anti-viral and antitumor immune responses can enhance the success of OVT."
Suggested Improvement: "A robust antiviral immune response can eliminate the virus and induce inflammation. Therefore, achieving a delicate balance between antiviral and antitumor immune responses is pivotal for enhancing the success of OVT."
Line 429:
Original: "Understanding the tropism of OVs to different cell types is vital in addressing these issues."
Suggested Improvement: "Comprehending the tropism of OVs for various cell types is essential to tackle these challenges."
Line 431:
Original: "Unlike many other cancer therapies, OVs have the potential to be used against many types of cancer malignancies and can be further combined with existing treatments, such as immunotherapy and cell therapy."
Suggested Improvement: "In contrast to numerous other cancer therapies, OVs possess the potential to combat a wide range of cancer malignancies and can be effectively integrated with existing treatments, including immunotherapy and cell therapy."
Line 433:
Original: "Combining OVT with other therapies might help overcome the limitations of those therapies."
Suggested Improvement: "The integration of OVT with other therapies has the potential to overcome the limitations associated with those treatments."
Line 434:
Original: "Thus, understanding the tropism of OVs in the complex TME is crucial for the success of developing combinatorial cancer therapy."
Suggested Improvement: "Hence, comprehending the tropism of OVs within the complex tumor microenvironment (TME) is indispensable for the successful development of combinational cancer therapies."
Author Response
This manuscript offers an extensive literature review on the utilization of DNA viruses in oncolytic virotherapy development. It compiles and elaborates on numerous virus modifications aimed at enhancing the selectivity of virus tropism toward cancer cells through diverse strategies. The text well represents the many scientific publications in the field, it is logical and well structured. However, both the text and the illustration of the review could be improved. Having an illustration in a review is always a winning strategy. I liked the idea of presenting three different viral families with three representatives and their schematic images. However, the arrow from the center leading to the different variants of genetic modifications confuses the reader. It gives the impression that all the examples refer to a central virus located in the center: Herpes Simplex virus 1. However, this is a false sense, two of the viral examples Myxoma virus and Vaccinia belong to the poxviridae family, representatives of which on the left. I would like the authors to revise the figure so that the viral schemes in the top row and the examples of oncolytic viruses or their modified variants at the bottom coincide. Another option is to separate the top and bottom levels of the figure with a horizontal line. Do this in a way that does not give the impression that all examples relate only to the central viral image. I would also like to correct the title of the illustration. More appropriate title: “Oncolytic DNA virus tropism modifications” or “Examples of oncolytic DNA virus tropism modifications”.
Several examples of virus classifications are located throughout the text of the review. It would be a good idea to illustrate such textual classifications with diagrams. In this case, illustrations improve and speed up the reader's perception of the information.
In the attached file - 39 comments to the text are given directly in pdf format and these comments are the most essential. The following section on the quality of the English language suggests minor changes to the text.
We thank the reviewer for all the comments. We believe that we have addressed all the comments raised by the reviewer in the revised manuscript.
We now modified the figure, and title in the revised manuscript.
We added all the suggested edits in the revised manuscript that are listed below. Please see the manuscript in TC.
Reviewer 2 Report
Comments and Suggestions for Authors
The authors declare a review to encompass the tropism of oncolytic DNA virus and how mutations/attenuations/engineering will endow OVs to tumors. However, there is very little specific information given on the cell pathways disrupted or how exactly the mutations in the OVs yields tropism. More infor on the normal viral lifecycle (with specific protein pathways/diagrams) should be generated. Additionally, the information in the article is mostly generalized, and should get into the weeds when describing the specific pox or HSV viral protein:host protein interactions which mediate tropism.
On a separate note, the OV field has generally moved towards an immunotherapy focus - the recruitment or bolstering of an adaptive immune response tailored towards the tumor. This omission in this review should be stated as the tropism of the virus is becoming less and less concerning for those working in the OV field.
The authors must go over their work and correct the grammatical and spelling errors, as well as double check their specific claims. Additionally, work to the table should be done in order to make it present a more clearer purpose. Is it a table organizing which genes in OVs pertain to host permissiveness/tropism or a general gallery of genes mutated in OVs?

English quality is OK but there are several errors in the manuscript. I tried to highlight them as they were found (attached manuscript with comments).
Author Response
The authors declare a review to encompass the tropism of oncolytic DNA virus and how mutations/attenuations/engineering will endow OVs to tumors. However, there is very little specific information given on the cell pathways disrupted or how exactly the mutations in the OVs yields tropism. More infor on the normal viral lifecycle (with specific protein pathways/diagrams) should be generated. Additionally, the information in the article is mostly generalized, and should get into the weeds when describing the specific pox or HSV viral protein:host protein interactions which mediate tropism.
We thank the reviewer for all the comments.
We thank the reviewer for this suggestion. In this review we have not focused much on the normal viral lifecycle, since there are several reviews already published on these viruses. Adding these new sections will also increase the length of this review article.
On a separate note, the OV field has generally moved towards an immunotherapy focus - the recruitment or bolstering of an adaptive immune response tailored towards the tumor. This omission in this review should be stated as the tropism of the virus is becoming less and less concerning for those working in the OV field.
We thank the reviewer for this suggestion. We also agree that OV filed is moving towards an immunotherapy focus. In this review, we tried to highlight the importance of understanding the tropism of these viruses and how engineering helped to enhance tumor tropism. How engineering OVs enhancing adaptive immune responses could be a topic for a future review.
The authors must go over their work and correct the grammatical and spelling errors, as well as double check their specific claims. Additionally, work to the table should be done in order to make it present a more clearer purpose. Is it a table organizing which genes in OVs pertain to host permissiveness/tropism or a general gallery of genes mutated in OVs?
We added all the suggested edits in the revised manuscript. We have updated the table. Please see the manuscript in TC.
Reviewer 3 Report
Comments and Suggestions for Authors
The authors have reviewed and summarized the natural and genetically engineered tumor tropism focusing on the most extensively studied DNA viruses for oncolytic virotherapy.
The review is well organized with clear flow and a thorough literature review, providing informative and up to date summary of the field.
Author Response
We thank the reviewer for all the positive comments.